# An integrated view on society readiness and initial reaction to COVID–19: A study across European countries

**Dalibor Petrović**[1,2]*, **Marijana Petrović**[2], **Nataša Bojković**[2], **Vladan P. Čokić**[3]

**1** Faculty of Transport and Traffic Engineering, University of Belgrade, Belgrade, Serbia, **2** Faculty of Philosophy, University of Belgrade, Belgrade, Serbia, **3** Institute for Medical Research, University of Belgrade, Belgrade, Serbia

* dalibor.petrovic@f.bg.ac.rs

**Data Availability Statement:** All relevant data are within the manuscript.

**Funding:** The author(s) received no specific funding for this work.

## Abstract

With the wake of the COVID-19 pandemic, the question of society's capability to deal with an acute health crisis is, once again, brought to the forefront. In the core is the need to broaden the perspective on the determinants of a country's ability to cope with the spread of the virus. This paper is about bringing together diverse aspects of readiness and initial reaction to a COVID-19 outbreak. We proposed an integrated evaluation framework which encapsulates six dimensions of readiness and initial reaction. Using a specific multi-level outranking method, we analysed how these dimensions affect the relative positioning of European countries in the early stages of the COVID-19 outbreak. The results revealed that the order of countries based on our six-dimensional assessment framework is significantly reminiscent of the actual positioning of countries in terms of COVID-19 morbidity and mortality in the initial phase of the pandemic. Our findings confirm that only when a country's readiness is complemented by an appropriate societal reaction we can expect a less severe outcome. Moreover, our study revealed different patterns of performance between former communist Eastern European and Western European countries.

## Introduction

A major health crisis, such as the COVID–19 pandemic declared by WHO on March 11 2020, is a huge challenge for any society. Such events become even more dangerous if the society itself is in a state of political, cultural or social crisis. This means that public health capacity to respond to such serious threats are significantly limited. Since government measures can have broad societal impacts, it is necessary that members of society comply with them as much as possible [1]. Therefore, pandemics are not only health issues but also political, economic and cultural matters. As elaborated in Lofredo [2] such circumstances can trigger waves of social confusion and panic, uncontrolled waves of contagion, and political and social unrest. Protests in the summer of 2020 against the government's restriction of coronavirus in Germany, Serbia, Romania, Bulgaria and Israel confirm these fears.

**Competing interests:** The authors have declared that no competing interests exist.

As the number of cases grows, the health system losses its resilience which calls for urgent limitation of the spread of the virus during the early stages of a pandemic. In such a situation, government interventions to support physical distancing take a crucial role in mitigating the severity of the outbreak [3–5]. This is especially important in the early days of a pandemic when additional capacity still needs to be provided and coping strategies revised. However, it is necessary to broaden the perspective on what makes society capable to manage such an acute health crisis. In line with this, several studies dealing with preparedness and vulnerability in the context of COVID-19 have emphasised the importance of social factors [6–8]. Our study is nested in this research domain and adds to previous findings by offering an integrated view on countries' readiness and reaction to the initial outbreak of COVID-19 in Europe. We use the term initial outbreak to highlight that we focused on the first four months of the outbreak (February, March, April and May 2020), given that during this period government interventions in European countries reached its peak and then started to decline.

Instead of focusing on the impact of a particular factor (e.g. social distancing) we gather different aspects of readiness and initial reaction to a pandemic. In order to be able to perform a cross-country analysis on this topic, we have transformed these aspects into a set of measurable dimensions. Our quantitative assessments were compiled based on a prominent evaluation framework for public health emergency preparedness and review of existing COVID-19 related impact studies. To make our conceptual framework operational, we combined well-established country-level indicators and depicted some specific variables associated with COVID-19 on the basis of recently established daily trackers. We ended up with six dimensions. Three of them are about the readiness of the society to fight the outbreak of the virus, while the rest reflect its initial reaction to COVID-19. Finally, we used the multi-level outranking method to investigate the extent to which these dimensions reflect the severity of the initial outbreak of COVID-19 in 23 European countries. This allowed us to rigorously analyse how the European countries stand relative to each other in terms of readiness and reaction to COVID-19.

This paper raises several research questions. The first is whether a better prepared health system of the country and trustworthy institutions could reflect the real outcome. The second is whether the application of more rigorous measures can complement the preparedness capacity of a country. The third question of interest concerns the existence of performance patterns across European countries.

## Materials and methods

### The conceptual framework

The conceptual basis of our study is inspired by the classic framework "structure-process-outcome (SPO)" [9] which was previously applied in the context of public health preparedness in emergency situations [10]. Based on the SPO framework and the findings of previous studies, we presented the outline of multidimensional analysis as given in Table 1. By **structure** we refer to the structural elements that existed before the pandemic and were crucial in determining society's readiness to cope with the pandemic. In the first raw, we draw attention to the health system and its preparedness for a health crisis, but also to the ability of society to cope with such events, as well as to the health risk factors that are present in a certain country. In turn, by **process** we refer to aspects of the reaction to the pandemic in order to reduce the virus transmission and mortality. In addition to government interventions and testing policies that have been recognized as instruments of the country's reaction to COVID-19, we have also included behaviour change as an important element of society reaction to the crisis. Finally, by **outcome** we refer to the severity of the COVID-19 outbreak in the initial phase.

**Table 1. Framework for multidimensional assessment of country readiness and reaction to COVID-19 outbreak.**

|  | *Institutional aspects* | *Medical aspects* | *Societal aspects* |
|---|---|---|---|
| *Readiness structure* | Preparedness capacity | Health risk factors | Trust in institutions |
| *Reaction process* | Government interventions | Testing policy | Mobility reduction |
| *Outcome* | Severity of initial COVID-19 outbreak | | |

The importance of outcome within the SPO is to show whether the system has achieved intended impacts on morbidity and mortality. Since the emergencies in public health are rare, there are few opportunities to include all three elements in assessment of society's response to the crisis [10]. Also, it takes time to see the whole picture, leaving researchers with narrowed insights while the pandemic is still on the run. Therefore, we chose to access societal readiness and reaction in the initial phase. This gave us the opportunity to connect the outcome of the pandemic (in terms of morbidity and mortality) together with readiness structure and reaction process.

As we can see from Table 1, the readiness structure and the reaction process include three aspects each, leading to a total of six dimensions. The proposed framework includes both a macro and micro perspective. Institutional aspects are related to the preparedness and reaction of institutions and as such belong to the analysis at the macro level analysis. Societal aspects indicate the behaviour and perception of individuals and they are a phenomenon at the micro level. In turn, medical factors are related to institutions and their policies, but they are also related to the health conditions of individuals.

The choice of relevant dimensions for this study was guided primarily by what we know so far. In this regard, we decided to include those components whose effects have been confirmed in numerous studies on COVID-19. As our research interest was European countries, some aspects such as economic, political or technological are not explicitly included. Some of them partially overlap with the included dimensions (economical as part of preparedness, democracy in high correlation with trust), while some do not have sufficient power of differentiation (such as the similar uptake of information and communication technologies (ICT) across Europe).

The literature is growing, but it is still limited when it comes to the influence of medical and non-medical factors on the spread of COVID-19. In the spotlight are government interventions [11] the preparedness of the healthcare system [12], risk factors [13], economic [14], political [15] and social factors. Among social factors, the one that received the most attention in the context of COVID-19 is the phenomenon of social distancing [16, 17]. Several other authors have focused on social trust [8], social behaviour [18] and the well-being of individuals [19]. Some other non-medical aspects, such as informational overload [20] and the role of ICT [21] also received considerable attention, but their significance has yet to be assessed.

We now proceed to elaborate on the relevance associated with the structure and process dimensions.

**Dimensions of the readiness structure–preparedness, trust and health risks.** The main question in a pandemic is what makes a society capable of dealing with such an acute crisis. Answers are usually built around the concept of **preparedness** and response to virus outbreaks [22]. Although generally considered to be the responsibility of the health system to respond to a crisis, there is a greater understanding that preparedness requires the involvement not only of the health sector but of society as a whole [23]. In a broader sense, health emergency preparedness can be seen as a capability of the public health and health systems, communities, and individuals to prevent, protect against, quickly respond to, and recover from emergencies,

particularly those whose scale, timing, or unpredictability threatens to overwhelm routine capabilities [22].

The outbreak of COVID-19 called in question the ability of well-prepared health systems to respond to the epidemic [24]. But the events of March and April 2020 in Italy and Spain reflected that the high level of preparedness was just one piece of the COVID-19 puzzle that should be followed by other compatible pieces, if the puzzle was to be solved.

The willingness of the public to comply to the measure, as proposed by the government, may be crucial for controlling the spread of COVID-19 and accordingly, personal rather than government action could be the most important issue [25]. Therefore, one of the key social elements for the successful implementation of government measures in a state of health crisis is the level of citizens' **trust** in the government and its health system. As pointed out by Legido-Quigley et al. [24] "the trust of patients, healthcare professionals, and society as a whole is of paramount importance for meeting health crises". Trust is an important factor because it can influence the public's judgments of risks and benefits. Decreased trust in the government's ability to deal with the threat can result in scepticism about public health warnings. The great health crisis of the past has confirmed that without trust in the government, the chances of society overcoming this type of crisis are significantly reduced [3]. Furthermore, trust may indirectly affect the acceptance of implemented measures, as people with higher levels of confidence in institutions are more likely to accept the recommended measures than those with lower confidence levels [26]. Studies related to the influenza A (H1N1) pandemic have already proven that the level of government trust and risk perception of the general public have been of great importance for the adoption of protective measures [3]. Similar conclusions can be found in early studies on COVID–19 [8, 27]. However, the link between trust in government and citizens' responses to government interventions is not as simple as might be expected. There is some evidence that there could be a positive relationship between low social confidence in government and a strong, determined government response to the pandemic [8].

In addition to preparedness and trust, another important element of the country readiness is the **health** condition of the population. The literature on health factors associated with COVID-19 is already extensive and covers many different medical problems. Special attention has been paid to various health risk factors that are fertile ground for COVID-19 eruption and spread, such as gender [28], age [13, 29], obesity and race [30], and comorbidities [31]. One of the earliest studies on risk factors for COVID-19 mortality was conducted at the epicentre of the COVID-19 outbreak in Wuhan and included 191 patients [13]. The results of this study showed an increasing chance of hospital death associated with older age. In another study [29], older patients (age ≥64 years) had higher mortality than younger patients (age ≤63 years) (36% vs. 15%; difference). Moreover, the increased risk of hospital death was over the age of 65 (mortality of 10.0%, vs. 4.9% among those aged ≤65) and current smoking [32]. Mortality rates for those who received mechanical ventilation in the age groups 18 to 65 and older than 65 years were 76.4% and 97.2%, respectively [31]. The overall case fatality rate was much higher among the elderly in China, Korea, and Italy [33]. Several statistical sources show that in European countries the hospitalized and deceased patients were elderly people (France: the mean age of hospitalized patients was 68 years and the mean age of the deceased was 79, Italy: 52.7% of COVID-19 cases were >60 years, while 95.4% of the deaths were people aged 60 and over [34, 35], UK: 40% of COVID-19 cases were aged 60 and over; 94% of the victims were people aged 70 and over [36, 37], Spain: 68.5% of hospitalized patients with COVID-19 were >60 years old, while 86.4% of deaths were people aged 80 and over [38, 39]. The data presented on COVID-19 of this large European population support aging as a major risk factor in both morbidity and mortality.

**Dimensions of the reaction process-government interventions, testing and mobility reduction.** Timely implementation of control measures is the key to their success, but it is also important to find a balance between early enough implementation to reduce the peak of the epidemic ensuring that they can be sustained in a timely manner [40]. It is fair to say that the rapid and strong reaction of the Chinese government and its success in fighting the virus in February 2020 influenced many other countries in terms of their strategies related to the outbreak of COVID-19. What has happened in China has shown that quarantine, social distancing, and isolation of the infected population can contain an epidemic [25]. Today, in response to COVID-19, many governments use a combination of containment and mitigation activities with the intention of delaying large patient surges and levelling the demand for hospital beds, while protecting the most vulnerable from infection, including the elderly and those with comorbidities [41]. This includes different ranges of contact tracing and self-isolation or quarantine but also varying levels of social distancing and promotion of public health measures. However, it is not easy to detect the individual effect of each **government intervention**. For example, one study finds that only the effect of lockdown is identifiable, and that it had a substantial effect of 81% reduction in virus transmission [42].

Along with government interventions, extensive **testing** may also play a major role in the fight against COVID-19. In his introductory speech at a press briefing on COVID-19 in the early days of raging pandemics, the WHO Director General Dr Tedros said: "We have a simple message for all countries: test, test, test" [43]. Mass testing leads to rapid case identification, rapid treatment of these people, and immediate isolation to prevent the spread of the virus [44]. Further, mass and timely testing is crucial to identify people who have come in contact with infected sources so they too can be treated quickly. In this way, it is possible to avoid the need for non-selective quarantines. Piguillem and Shui [45] argue that random testing can even replace the lockdown policy and eliminate the need for nonselective quarantine. Regular screening regardless of symptoms has also been analysed by Grassly et al. [46], who claim that if performed for health care and other key workers, it can prevent a third of transmissions and greatly complement lockdown interventions.

Studies conducted in China [47, 48] and the United States [49] have provided early evidence that social distancing can successfully reduce infection transmission. **Reduction of mobility** is at the heart of social distancing [17]. It is important that the decision to stay at home is not only the result of an imposed government restriction, but also reflects the self-imposed COVID-19 strategies of avoiding individuals [16, 50, 51]. Several early studies have shown that citizens stay at home voluntarily, rather than in response to stay-at-home orders. Mehari [8] found that the voluntary choice to stay at home was associated to the spread of news about the first COVID-19 case. Engle et al. [52], report that an increase in the local infection rate from 0% to 0.003% reduces mobility by 2.31%, while a government restriction order to stay-at-home reduces mobility by 7.87%. A study from the United States [53] found that every resident, even without government mandates, was able to help slow the spread of COVID-19. Espinoza et al. [54], also argue that individual patterns of mobility established by people may be sufficient to contain epidemics.

## Methodological approach

Based on established conceptual framework we developed methodological approach which includes four steps. The first one is on operationalisation of dimensions and entails two sub-steps: 1) establishing indicators and 2) depicting variables for each indicator The step that follows is on sampling and extracting data for each variable. In the third step we have performed a pre-scanning analysis of data based on correlation, analysis. Finally, in the fourth step we

**Table 2. Operationalisation of dimensions—Summary.**

| | Dimensions | Explanation in brief | Operationalization | | |
|---|---|---|---|---|---|
| | | | Indicator(s) with label in brackets | Variable | Data Source |
| 1. | **Preparedness** | Country's capacity to deal with the importation and the spread of the virus | IHR compliance (IHRc) | Average value of SPAR indicators for following capacities: Legislation and Financing, IHR Coordination, Laboratory Surveillance, Human Resources, NH Emergency Framework, Health Service Provision, Risk Communication, Points of Entry | eSPAR (WHO) [55] |
| 2. | **Trust** | Confidence in institutions to deal with health crisis | Trust in government (TrGov) | Level of trust in government on scale 1–4* | EVS [56] |
| | | | Trust in health care system (TrPHS) | Level of trust in public health system on scale 1–4* | |
| 3. | **Health risk factors** | Health risk factors that affect vulnerability of citizens to COVID-19 | Share of older people (HRF) | Percentage of population aged 65 or more | Our World in Data [57] |
| 4. | **Government interventions** | Stringency of government reaction—school closures; workplace closures; cancellation of public events; restrictions on public gatherings; closures of public transport; stay-at-home requirements; public information campaigns; restrictions on internal movements; and international travel controls. | Timeliness of government interventions (GovI1) | Number of days from the first case to the maximum Stringency index value | Stringency Index -OxCGRT -Oxford COVID-19 Government Response Tracker [58] |
| | | | Strictness of government interventions (GovI2) | Average Stringency Index for days when its score was over 60 | |
| | | | Duration of government interventions (GovI3) | Number of days before or after the first case when Stringency Index was at least 10 | |
| 5. | **Testing policy** | Scale of testing compared to the scale of the outbreak | Mass testing (Test) | Number of days with less than 10 tests per confirmed case | Our World in Data [59] |
| 6. | **Behavioural change** | Mobility reduction as proxy | Strength of mobility reduction (MobR1) | Average of all <40% mobility decrease values | Apple Community Tracker (data stream 'walking') [60] |
| | | | Promptness of mobility reduction (MobR2) | Number of days after the first case when mobility drops below 40% | |

*Likert scale, re-coded in reverse order ("none at all" takes value of 4, with "a great deal" takes value of 1).

have conducted a cross-country analysis using a specific multiple criteria decision making method. Besides data pre-processing and calculations, this final step includes customized visualization of country positioning. Each step is presented in separate subsection.

**Operationalisation of dimensions.** In this subsection, we define indicators, variables and data sources for the operationalisation of the dimensions. The proposed approach is summarised in Table 2 and further elaborated for each dimension.

**Preparedness** is operationalised on the basis of the e-SPAR database, which contains assessments of the country's capacity to meet IHR—International Health Regulations. e-SPAR stands for Electronic State Parties Self-Assessment Annual Reporting Tool and provides data on global public health security at the country level. It has been used in several studies to provide insight into countries' preparedness for the COVID-19 outbreak [12, 61]. The 2019 version of e-Spar contains 13 capacity ratings. Each capacity is measured by one to three indicators, which makes to a total of 24 indicators. Similar to Gilbert et al. [12] we excluded capacities for zoonoses, food safety, chemical events, and radiation emergency.

**Trust** is operationalised around two standpoints: *holding a positive perception might increase the efficiency and effectiveness of government operations* [62] and *sense of confidence in the health system will lead people to comply with the recommendations* [63, 64]. Accordingly, we have included two pillars of vertical trust—trust in the government and trust in the public health system; both quantified using data from the last wave of the European Values Study [56].

As we opted for age as a **health risk factor**, we used data on the proportion of people aged 65 or over. As previously elaborated, at this stage of medical research it is difficult to establish unambiguous conclusions on COVID-19 health risk factors. That is why we decided to use the one that has been proved to be most directly associated with the severity of COVID-19.

In order to operationalise **government interventions**, our study uses a composite measure that includes eight indicators from the category containment and closure into the so-called Stringency index (SI). SI is part of the Oxford COVID-19 Government Response Tracker [65], a daily updated database on composite measures covering three categories: containment and closure, the economy and the health system. This tracker allowed researchers to extend the analysis to a larger set of countries and explore the link between government measures and other aspects (such as social distancing in several studies [15, 66, 67]). To expand the daily tracker into an indicator of government intervention, we depicted three variables. As we focus on the initial reaction, these three variables (Table 2) represent the timeliness, strictness and duration of government interventions within a defined time frame (from the first case in each country to June 1). The standpoint behind this is that interventions should be implemented early enough, strict enough and long enough.

When it comes to the **testing policy**, this study does not address who was tested, but only deals with the number of tests performed in each country. Data on the scope of testing are scarce and difficult to obtain systematically. A data source that can be helpful is Our World in Data. This open-source platform contains data on the scale of testing in relation to the scale of the outbreak. It allowed us to build our indicator around WHO recommendation that proposes about 10–30 tests per confirmed case as a general benchmark of adequate testing [68]. In respect to the nature of the available data, we opted for the number of days during the initial outbreak, in which testing was below the suggested benchmark.

Apple's Mobility Trends Reports were used to quantify the degree of **mobility reduction** in European countries during the initial phase of the COVID-19 outbreak. Similar to the approach we used in the case of government interventions, two variables are created to reflect strength and promptness of mobility change relative to the first case in each country (Table 2).

**Sampling and data.**  Regarding the country sample our aim was to collect data for at least 10 Eastern European and 10 Western European countries, depending on data availability. After exploring data sources for the selected indicators and variables, we were able to perform our analysis for 23 European countries (12 Western European and 11 Eastern European) (Table 3). Data on structure (IHR compliance, trust, and risk factors) are the latest available, while data on reaction process (government interventions, testing, and mobility reduction) refer to the period of the initial response to COVID-19 (February, March, April, and May 2020).

**Correlation among dimensions.**  The correlation, matrix in Table 4 indicates several interesting relationships between dimensions. As expected, there is a positive correlation between government interventions (GI) and several other dimensions. The strongest relationship is between timeliness of GI and promptness of mobility reduction (MR). There is also a strong correlation between duration of GI and promptness of MR and strictness of GI with the strength of MR. This tells us that the MR is a direct consequence of government interventions. However, we can also notice that the promptness of MR is in a strong negative correlation

**Table 3. Data for 23 European countries.**

|  | IHRc | TrGov | TrPHS | HRF | GovI1 | GovI2 | GovI3 | Test | MobR1 | MobR2 |
|---|---|---|---|---|---|---|---|---|---|---|
| Desired direction | ↑ | ↓ | ↓ | ↓ | ↓ | ↑ | ↓ | ↓ | ↓ | ↓ |
| AUT | 71.44 | 2.71 | 1.92 | 19.2 | 20 | 76.02 | -2 | 21 | 30.9 | 19 |
| BGR | 62 | 3.02 | 2.98 | 20.08 | 14 | 69.9 | -14 | 0 | 29.08 | 8 |
| CHE | 95 | 2.26 | 2.19 | 18.4 | 21 | 73.6 | 2 | 25 | 34 | 26 |
| CZE | 65 | 3.1 | 2.3 | 19 | 22 | 73.7 | -36 | 0 | 26.7 | 14 |
| DEU | 88.7 | 2.74 | 2.34 | 21.4 | 55 | 72.1 | 16 | 1 | 40+ | 62 |
| DNK | 97.3 | 2.69 | 2.05 | 19.7 | 35 | 72.9 | 1 | 6 | 40+ | never |
| ESP | 85.9 | 3.12 | 2.02 | 19.4 | 59 | 80.9 | -1 | 40 | 14.7 | 43 |
| EST | 72.3 | 2.61 | 2.33 | 19.4 | 31 | 72.86 | 14 | 4 | 40+ | 32 |
| FIN | 92 | 2.63 | 1.97 | 21.2 | 58 | 66 | 7 | 1 | 40+ | 74 |
| FRA | 78.6 | 2.97 | 1.96 | 19.7 | 53 | 87.9 | -1 | 46 | 17.68 | 51 |
| GBR | 93 | 2.86 | 1.84 | 18.5 | 56 | 73.5 | 3 | 42 | 37 | 55 |
| HRV | 77.3 | 3.39 | 2.72 | 19.7 | 27 | 93.2 | -27 | 13 | 28.13 | 25 |
| HUN | 66.9 | 2.8 | 2.76 | 18.6 | 24 | 70.4 | -5 | 0 | 30.7 | 14 |
| ITA | 88 | 2.99 | 2.37 | 23 | 73 | 81.4 | 1 | 38 | 20.4 | 39 |
| LTU | 80 | 2.66 | 2.54 | 19 | 18 | 78.9 | 1 | 0 | 35.15 | 24 |
| NLD | 91.2 | 2.56 | 2.12 | 18.8 | 33 | 75.6 | 11 | 43 | 37.4 | 25 |
| NOR | 94.7 | 2.37 | 1.83 | 16.8 | 27 | 69.7 | -28 | 0 | 40+ | never |
| POL | 76.33 | 3.02 | 2.58 | 16.8 | 37 | 83 | -7 | 0 | 27.36 | 12 |
| ROU | 65 | 3.21 | 2.58 | 17.8 | 34 | 80.7 | -3 | 12 | 25.4 | 20 |
| SRB | 75.89 | 3.04 | 2.73 | 20 | 16 | 96.7 | -11 | 41 | 26.7 | 11 |
| SVK | 77.3 | 2.86 | 2.46 | 15.1 | 33 | 74.9 | -4 | 0 | 34.4 | 10 |
| SVN | 84.4 | 3.1 | 2.54 | 19.1 | 26 | 82.5 | -1 | 0 | 33.1 | 16 |
| SWE | 93.3 | 2.5 | 1.98 | 20 | 84 | 84 | 20 | 60 | 40+ | never |
| AVG | 81.37 | 2.84 | 2.31 | 19.16 | 37.22 | 74.19 | -2.78 | 37.22 | 28.75 | 29.00 |

Indicators: IHRc—IHR compliance; TrGov—Trust in government; TrPHS—Trust in health care system; HRF—Share of older people; GovI1—Timeliness of government interventions; GovI2—Strictness of government interventions; GovI3—Duration of government interventions; Test—Mass testing; MobR1—Strength of mobility reduction; MobR2—Promptness of mobility reduction

with trust in the health care system. This means that countries with higher level of trust have not reduced their mobility as much as those countries with lower level of trust. Therefore, it is fair to say that mobility reduction is not only a consequence of government interventions, as we discussed above, but also a spontaneous reduction of citizen's mobility as a personal precaution (in the context of their low trust in the health care system).

It can further be observed that the timeliness of GI is strongly negative correlated with country's preparedness (IHRc) and with trust in the health system (TrPHS), meaning that countries with lower preparedness and lower trust have implemented measures more promptly. The strength and duration of the GI are also negatively correlated with trust in the government which supports the assumption that government with lower trust reacted quickly and strictly because they feared that their citizens would not comply with the imposed measures. Both trust in the government and in the public health system are strongly positively correlated with the IHRc. In other words, countries with better preparedness have a more trustworthy government and the health care system.

**A method for cross-country analysis.** Our study proceeds with fusing different aspects of countries' preparedness and response to COVID-19 to gain an integrated view of their performance. We use a multiple-criteria decision making approach (MCDA) based on outranking

**Table 4. Correlation matrix.**

| | | GovI1 | GovI2 | GovI3 | IHRc | TrGov | TrPHS | HRF | Test | MobR1 | MobR2 |
|---|---|---|---|---|---|---|---|---|---|---|---|
| GovInt1 | Pearson Corr. | 1 | .502* | .530** | -.505* | -.059 | -.492* | .418* | .543** | -.054 | .790** |
| | Sig. (2-tailed) | | .015 | .009 | .014 | .788 | .017 | .047 | .007 | .808 | .000 |
| | N | 23 | 23 | 23 | 23 | 23 | 23 | 23 | 23 | 23 | 20 |
| GovInt2 | Pearson Corr. | .502* | 1 | .416* | -.286 | -.488* | .327 | -.068 | -.255 | .460* | .246 |
| | Sig. (2-tailed) | .015 | | .049 | .187 | .018 | .127 | .758 | .240 | .027 | .295 |
| | N | 23 | 23 | 23 | 23 | 23 | 23 | 23 | 23 | 23 | 20 |
| GovInt3 | Pearson Corr. | .530** | -.416* | 1 | .409 | -.430* | -.277 | .289 | .350 | .340 | .525* |
| | Sig. (2-tailed) | .009 | .049 | | .053 | .040 | .201 | .181 | .101 | .113 | .017 |
| | N | 23 | 23 | 23 | 23 | 23 | 23 | 23 | 23 | 23 | 20 |
| IHRc | Pearson Corr. | .505* | -.286 | .409 | 1 | .553** | .656** | .161 | .347 | -.421* | -.638** |
| | Sig. (2-tailed) | .014 | .187 | .053 | | .006 | .001 | .463 | .105 | .045 | .002 |
| | N | 23 | 23 | 23 | 23 | 23 | 23 | 23 | 23 | 23 | 20 |
| TrGov | Pearson Corr. | -.059 | .488* | -.430* | -.553** | 1 | .520* | .088 | -.052 | -.678** | -.236 |
| | Sig. (2-tailed) | .788 | .018 | .040 | .006 | | .011 | .688 | .814 | .000 | .316 |
| | N | 23 | 23 | 23 | 23 | 23 | 23 | 23 | 23 | 23 | 20 |
| TrPHS | Pearson Corr. | -.492* | .327 | -.277 | -.656** | .520* | 1 | -.017 | -.411 | -.232 | -.659** |
| | Sig. (2-tailed) | .017 | .127 | .201 | .001 | .011 | | .938 | .051 | .287 | .002 |
| | N | 23 | 23 | 23 | 23 | 23 | 23 | 23 | 23 | 23 | 20 |
| RF | Pearson Corr. | .418* | -.068 | .289 | .161 | .088 | -.017 | 1 | .284 | -.123 | .524* |
| | Sig. (2-tailed) | .047 | .758 | .181 | .463 | .688 | .938 | | .189 | .577 | .018 |
| | N | 23 | 23 | 23 | 23 | 23 | 23 | 23 | 23 | 23 | 20 |
| Tst | Pearson Corr. | .543** | -.255 | .350 | .347 | -.052 | -.411 | .284 | 1 | -.294 | .266 |
| | Sig. (2-tailed) | .007 | .240 | .101 | .105 | .814 | .051 | .189 | | .174 | .256 |
| | N | 23 | 23 | 23 | 23 | 23 | 23 | 23 | 23 | 23 | 20 |
| MobR1 | Pearson Corr. | -.054 | -.460* | .340 | .421* | -.678** | -.232 | -.123 | -.294 | 1 | .161 |
| | Sig. (2-tailed) | .808 | .027 | .113 | .045 | .000 | .287 | .577 | .174 | | .498 |
| | N | 23 | 23 | 23 | 23 | 23 | 23 | 23 | 23 | 23 | 20 |
| MobR2 | Pearson Corr. | .790** | -.246 | .525* | .638** | -.236 | -.659** | .524* | .266 | .161 | 1 |
| | Sig. (2-tailed) | .000 | .295 | .017 | .002 | .316 | .002 | .018 | .256 | .498 | |
| | N | 20 | 20 | 20 | 20 | 20 | 20 | 20 | 20 | 20 | 20 |

*Correlation is significant at the 0.05 level (2-tailed).

**Correlation is significant at the 0.01 level (2-tailed).

relations. Within this area of MCDM we decided on a specific method and software solution, ELECTRE MLO developed in Petrovic et al. [69]. The method itself has been shown to be easily applicable for cross-country comparative reporting on various topics [70–75].

A common idea of the outranking method is to perform pair-wise comparisons and establish binary relations between items (alternatives) being observed. In the family of the ELECTRE outranking methods, these relations determine whether an item is at least as good as another, given that there may be arguments to reject such a claim. An item is better than another if it performs better on a sufficient majority of criteria, which is verified by the concordance threshold. At the same time, the outranking condition may not hold if there is strong opposition to this claim among the remaining minority of criteria. This is controlled by the discordance threshold.

Among the many upgrades of the ELECTRE method, ELECTRE MLO stands out as a method that arranges items into hierarchical performance levels. It introduces a specific

procedure by which preferable items are repeatedly extracted. After removing the best performers from the set, in the next iteration, the best among the others are searched, etc. As a result, the items are distributed according to the level of their performance which is visualised in the form of a so-called relation tree. A higher position (level) on a relation tree implies better performance.

In this study, we first arrange countries according to their structural readiness to cope with COVID-19. We then separately include response process indicators to outline the individual impacts of each. Eventually, we include all dimensions and analyse to what extent obtained performance levels correspond to the actual outcome.

In addition to the degree of matching of expected and achieved performance at the level of the entire set, a series of relation trees will outline useful information for each country. These data refer to the vertical mobility of countries—an increase to a higher or a decrease to a lower performance. To analyse this, we have created additional ELECTRE MLO output—visualization of the vertical mobility of countries.

In order to provide the data collected from the ELECTRE MLO (given in Table 3), it is necessary to transform it into performance scores. The scale from 1 to 9 is most often used, 1 is the lowest grade and 9 is the highest. In order to differentiate the topics and avoid interference among related indicators, each dimension is represented by one criterion and equally weighted. For this purpose, trust, government interventions and mobility related indicators are averaged. Concordance threshold is set to reflect that dominance is declared when the unit (here the country) exhibits at least the same performance by 70% of the criteria while and discordance threshold is set to q = 0,3.

Performance scoring is shown in Table 5.

In order to simultaneously monitor and compare the performance obtained by the ELECTRE MLO (expected performance) with the actual outcome (achieved performance), the countries are marked with colours (Table 6). We used three colours—red, yellow and green, each representing the severity level of the outcome (high, middle and low, retrospectively). The level of severity of outcomes is arbitrarily set as follows: countries with less than 1000 reported cases per million residents and less than 30 deaths per million residents are "green"; those with 1000–2000 reported cases and 30–100 deaths are "yellow", while countries with more than 2000 reported cases per million residents and more than 100 deaths per million residents are marked as "red". Data on the number of confirmed cases and deaths related to COVID-19 were extracted from the European Centre for Disease Prevention and Control and refer to cumulative data from the beginning of the COVID-19 pandemic until June 1, 2020 [76].

## Results

According to the ELECTRE MLO procedure five relation trees were obtained and presented on Figs 1 and 2. Each country is represented by its flag with a border line coloured according to the outcome of the outbreak at an early phase. The left relation tree in Fig 1 visualises the positioning of countries only on the basis of the tree readiness dimensions while the right is obtained on the basis of all six dimensions. The three relation trees in Fig 2 picture relative position of countries when only one reaction dimension is added to the three readiness dimensions: mobility reduction (relation tree a) in Fig 2), government interventions (relation tree b) in Fig 2) and testing policy (relation tree c) in Fig 2).

Relation trees allowed us to investigate how the inclusion of different dimensions affect the positioning of countries with different levels of outbreak severity in the early phase (green, yellow and red countries according to Table 6).

**Table 5. Performance scores.**

|       | IHRc | Trust | RskF | GovI | Test | MobRed |
|-------|------|-------|------|------|------|--------|
| AUT   | 7    | 6     | 5    | 6    | 6    | 7      |
| BGR   | 6    | 3     | 4    | 6    | 9    | 7      |
| CHE   | 9    | 6     | 6    | 6    | 6    | 6      |
| CZE   | 6    | 4     | 5    | 7    | 9    | 8      |
| DEU   | 8    | 5     | 2    | 3    | 9    | 2      |
| DNK   | 9    | 5     | 4    | 5    | 9    | 1      |
| ESP   | 8    | 5     | 5    | 5    | 3    | 8      |
| EST   | 7    | 5     | 5    | 4    | 9    | 4      |
| FIN   | 9    | 6     | 3    | 3    | 9    | 2      |
| FRA   | 8    | 5     | 4    | 5    | 3    | 7      |
| GBR   | 9    | 5     | 6    | 4    | 3    | 3      |
| HRV   | 7    | 3     | 4    | 8    | 8    | 7      |
| HUN   | 7    | 4     | 6    | 6    | 9    | 7      |
| ITA   | 8    | 4     | 1    | 4    | 4    | 7      |
| LTU   | 8    | 5     | 5    | 6    | 9    | 6      |
| NLD   | 9    | 5     | 5    | 5    | 3    | 5      |
| NOR   | 9    | 6     | 8    | 7    | 9    | 1      |
| POL   | 7    | 4     | 8    | 6    | 9    | 8      |
| ROU   | 6    | 4     | 6    | 6    | 8    | 8      |
| SRB   | 7    | 4     | 4    | 8    | 3    | 8      |
| SVK   | 7    | 5     | 9    | 5    | 9    | 6      |
| SVN   | 8    | 4     | 5    | 6    | 9    | 6      |
| SWE   | 9    | 6     | 4    | 1    | 1    | 1      |

As we noted the first relation tree in Fig 1 reflects the positioning of countries in terms of their readiness: preparedness, trust and the share of the elderly population. Countries with better functional capacities to deal with the acute health crisis, which have more trustworthy institutions (government and public health system) and a younger population are expected to be less sensitive to the outbreak of COVID-19. However, this first relation tree implies that this is not the case. Four of the eight countries that occupy the first three levels are red (Switzerland, Great Britain, Netherlands and Sweden). Although they had a comparative advantage in terms of their structural capacity, they had a more severe initial outcome. The same is true for the bottom levels, where three countries with a less severe outcome can be found (green Czech Republic, Croatia and Bulgaria). Slovakia (green) and Italy (red) are the only two countries in which the structural capacity to deal with an acute health crisis coincides with the severity of the initial outbreak of COVID-19.

We will now look at the effects of the reaction process criteria. When we introduce the criterion of mobility reduction, as expected, the red countries begin move downwards, while the green ones move in the opposite direction (Fig 2a). These changes are slow but evident. With exception of Poland all green countries have slightly improved their relative position. There are still four red countries on the first three levels. Spain is new here due to large reduction in mobility. Low performance by the same criteria 'pushed down' the Netherlands and Germany. The addition of government interventions affected the hierarchy more than the change in mobility. Only two red countries retained their positions (Great Britain and Spain), while the rest went down the relation tree. The same goes for the green countries. Due to quick and strict government measures, two managed to maintain their position while the others climbed up.

**Table 6. Countries in terms of severity of initial outbreak.**

|  | Cases per million inhabitants | Deaths per million inhabitants |
|---|---|---|
| SVK | 278.406 | 5.129 |
| BGR | 357.634 | 19.573 |
| HRV | 546.858 | 24.846 |
| LTU | 608.311 | 24.979 |
| POL | 603.092 | 27.427 |
| CZE | 853.489 | 29.788 |
| SRB | 1660.642 | 35.417 |
| NOR | 1549.645 | 43.532 |
| EST | 1395.36 | 49.754 |
| SVN | 708.537 | 51.95 |
| HUN | 397.605 | 53.518 |
| FIN | 1216.99 | 56.491 |
| ROU | 976.781 | 63.885 |
| AUT | 1836.805 | 74.169 |
| DNK | 1987.499 | 98.063 |
| DEU | 2153.849 | 100.855 |
| CHE | 3548.741 | 191.112 |
| NLD | 2681.666 | 344.502 |
| SWE | 3537.582 | 422.407 |
| FRA | 2283.79 | 439.106 |
| ITA | 3832.699 | 548.148 |
| GBR | 3964.396 | 557.361 |
| ESP | 5088.378 | 580.026 |

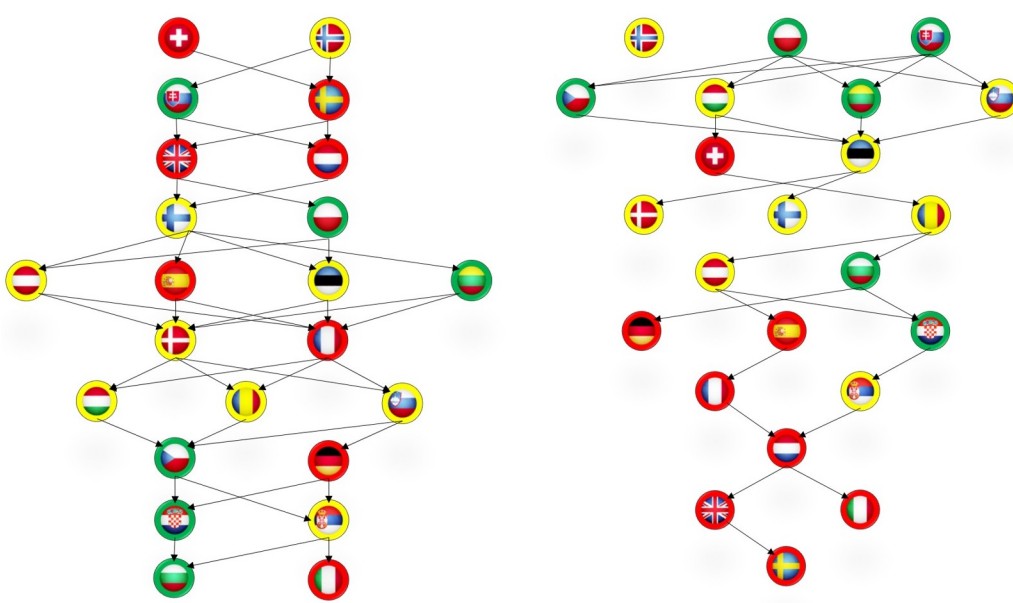

**Fig 1. Hierarchical order of countries based on readiness dimensions (left) and structure +reaction dimensions (right).**

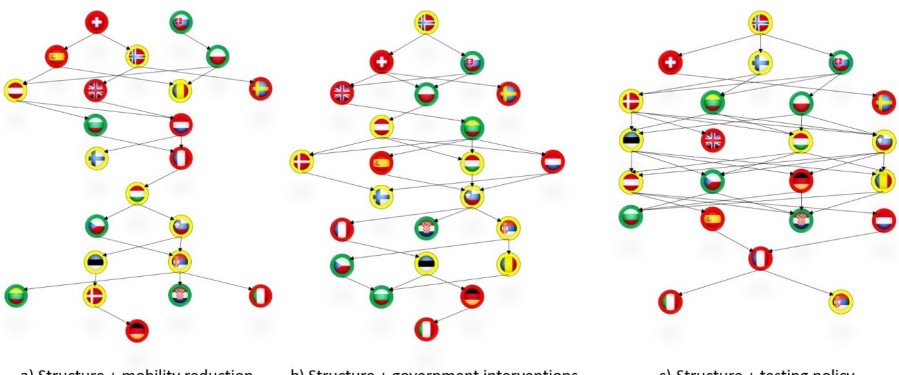

a) Structure + mobility reduction  b) Structure + government interventions  c) Structure + testing policy

**Fig 2. Hierarchical order of countries after adding different dimensions of the process.**

When testing is added (as the fourth dimension) number of levels is reduced from 10 to 8 which implies that it has smoothened differences in structural capacities (it has technically helped some countries to be released from the dominance of their counterparts). For example, this is the case with Poland which was dominated by Great Britain. Due to the significant number of days with testing below the proposed benchmark the red countries fell through the hierarchy. The exception is Germany which climbed tree because its scope of testing is significantly higher compared to many other countries. Again green countries have improved their relative position indicating that the testing criteria are in favour of these countries.

We now turn to the final result which signifies the cumulative effect of societal reaction aspects. When we include all six dimensions, the relation tree becomes most reflective in terms of the severity of the initial COVID-19 outcome. Namely, the positioning of countries in relation to the six dimensions is closer to the positioning in terms of the outbreak severity. As can be seen from the right graph in Fig 1 there are no red countries on the first two levels, nor green on the four bottom levels. Although we did not end up with a perfect 'reverse traffic light', the final hierarchy (Fig 2) corresponds well to the actual outcome.

To illustrate the changes in the positions of countries caused by the cumulative effect of different aspects of societal reaction we created a summary graph presented in Fig 3. The graph visualises 'vertical mobility' of countries from the initial position in relation to the three dimensions of readiness (dots on the graph) to the final position induced by all six dimensions (country flags). The higher end position on the graph reflects better country performance. Both dots and flags were coloured (in red, yellow and green) to reflect the actual outcome—the severity of COVID-19 at an early stage (Table 6).

Looking at this overall image, we notice that almost all countries have changed their placements on the relation tree. Green and yellow migrated upwards, while the reds slid downwards, thus confirming the simultaneous effect of readiness and reaction variables.

As for the green countries, some of them had an unfavourable starting position due to poorer readiness performance. Despite their unsatisfactory readiness for the epidemic, they managed to improve their position with a prominent societal reaction. This is especially true for Bulgaria, Croatia and the Czech Republic. Comparatively better placement was achieved by those countries that have more favourable initial conditions expressed by the assessment of structural variables, such as Poland and Slovakia.

When it comes to the red countries, we may say that they generally had a good predisposition to cope with COVID-19, but their reactive measures were relatively worse. If we look at green Bulgaria and red Italy as the countries with the lowest rank according to structure

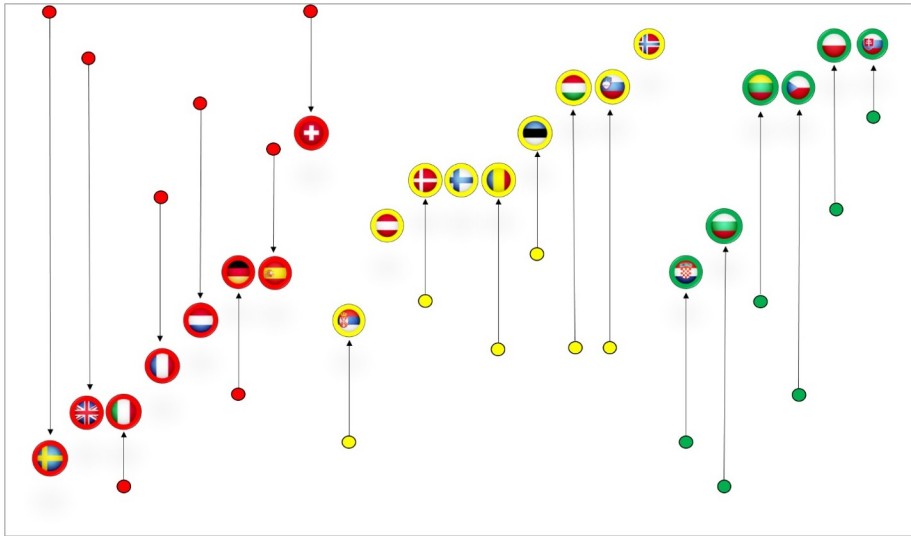

**Fig 3. Summary graph—Vertical mobility of countries induced by three reaction dimensions.**

dimensions—we can see that Bulgaria has managed to improve its relative position. Unlike Italy, strong government interventions and testing policy has allowed Bulgaria to move five levels higher. The most obvious drop among red countries came from Sweden and Great Britain. Both experienced a cumulative effect of all three reaction dimensions. Neither of the three reaction dimensions had the power to significantly reduce the relative position of these red countries (Fig 2). However, their joint contribution resulted in a significant slide in the hierarchy.

Germany is the only red country which managed to improve its position due to reaching a benchmark in testing. But this was not enough for Germany to move to the upper positions, which also corresponds to the real outcome.

Three (yellow) countries—Norway, Finland and Austria retained their positions, which implies that these countries had a balanced performance in terms of readiness and reaction to the pandemic.

Another observation can be made from the sum graph (Fig 3). All Eastern European countries have progressed thanks to rapid societal reaction. In contrast, almost all Western societies ended low in the final hierarchy, despite good readiness. The exception are Scandinavian countries. Excluding Sweden, they all managed to maintain their relative position.

## Discussion and conclusions

There is no doubt that the current situation regarding the outbreak of the COVID-19 is a multidimensional concern. Our study supports this view with an integrated view of impact factors with the focus on the initial outbreak in 23 European countries. At the heart of our analysis is the joint contribution of the six dimensions in dealing with the initial outbreak of a pandemic, which are divided into the structural readiness and the societal reaction. The structural readiness reflects the state in which society awaits the pandemic. It includes three dimensions: health system preparedness, trust in institutions, and health risk factors. The process of societal reaction to an outbreak is also characterized by three dimensions: government interventions, testing policy, and behavioural change. We came up with a set of indicators which reconcile several criteria: the essence of each dimension, specifics of the research area and the time

frame of the analysis, as well as the feasibility of data collecting. Our analysis is built on the idea that only when these different aspects come together we can move further to scrutinising the discrepancy between expected and achieved outcome (society's readiness/reaction vs. severity of the outcome in the early stage of a pandemic).

The outcome of the initial phase of the COVID-19 outbreak clearly showed that having a better prepared health system and trustworthy institutions is not enough to count on success in dealing with the spread of virus such as COVID-19. With the help of specific multiple-level outranking method, we came to the following main findings.

- The process of reaction to the outbreak to the virus was more important than the structural predispositions of the country. What was even more striking, societies with lower prepared-ness capacities and untrusted institutions performed better in the initial phase of the COVID-19 outbreak.

- Government interventions are considered to be the most influential. However, it alone can-not reflect the actual outcome. Only when other dimensions of societal reaction are included, the expected and achieved performance is fairly aligned.

- The differences between former communist Eastern European countries and Western Euro-pean countries in reaction to COVID-19 outbreak are evident.

Our findings reveal the full complexity of efforts to bring COVID-19 pandemic under con-trol. Although we have presented the most important dimensions of the societal reaction to COVID-19 they can only be understood in relation to other dimensions. For example, coun-tries with better prepared health systems are also countries with similar political traditions and cultural values. Therefore, it is not easy to explain why their reaction to COVID-19 was weaker compared to countries with different political and cultural background.

We can also question methodologies and metrics for assessing and comparing prepared-ness. In the era of globalisation and migration, country-level metrics may fail to reflect pre-paredness at the national-level [6]. What also rises the mist around e-SPAR and similar indicators (Global Health Security Index, Infectious Disease Vulnerability Index) is the rarity of epidemics, which in combination with the exclusive self-evaluation results, and neglect of demographic, socio-economic, and political factors may blur the reflection of a country's pre-paredness and vulnerability [6, 12, 77].

There are similar dilemmas when it comes to understanding government interventions. They are mainly focused on social distancing and their scope is actually a measure of the effec-tiveness of an uptake of the government reaction. However, this is not simple, because social distancing comes not only as a consequence of lockdown measures, but also as a spontaneous reaction of people to the news of the first case, as we discussed in opening sections.

If we recall our correlation analysis, we can see that the strength of government interven-tions is related to trust in institutions, but in the opposite direction than might have been expected. We found that countries with lower trust in institutions had more stringent and prompt interventions, followed by a better outcome in the initial phase of the outbreak. Our data showed (Table 3) that countries with less severe initial outbreaks had an average trust score of 2.6 (on a four step reverse scale) compared with a score 2.06 in countries with a more severe initial outbreak. It seems that less trustworthy governments had to count on quick lock-down interventions instead of relying on the conscience of citizens' (to follow recommenda-tions), which in turn could lead to a better result compared to more trustworthy countries. However, although low social trust plays a positive role in the initial months of the pandemic, it is unclear whether this will continue to be the case as the current situation evolves. Low social trust can lead to catastrophic effects when countries reopening is characterized by

uncertainty and lack of clear direction, as manifested in the US and some Balkan countries during the summer of 2020.

Another interesting finding is a comparison between Western European and Eastern European countries. In terms of structural readiness, Western countries were in a much better position, but the Eastern European countries had a better reaction to the outbreak. One of the highlighted reasons for this is swift lockdowns and bans evident mostly in central and Eastern European countries also known as former communist countries (Poland, Bulgaria, Hungary, the Czech Republic, Slovakia). The governments of these countries were aware of its weaknesses (vulnerable health system, lower initial testing capacities, and low trust in government) and had to implement early lockdown in order to prevent a pandemic eruption. According to Pancevski and Hinshaw [78], the main cause of this discrepancy is the fact that "the poorer countries of Central and Eastern Europe, fearing their relatively weak healthcare systems would be overwhelmed by the virus, moved more quickly to enact strict social-distancing rules and restrict movement to contain outbreaks".

Unlike Western European countries, Eastern countries began measures long before they reached the 1000th case. Perception of the weaknesses of health system and how trustworthy it is, have also made people in these countries feel more vulnerable and follow the lockdown measures.

However, a better overall preparedness score for Western health systems does not necessarily mean that they are better prepared to fight viruses like COVID-19. For example, when it comes to the number of hospital beds per 100 000 inhabitants, in 2018 the former communist countries ranked 8 out of the top 10 and 11 out of the top 15 in Europe [79]. This is important because about 1/5 of patients with COVID-19 who have moderate or severe disease require hospitalisation [80, 81]. Among hospitalised patients with COVID-19, the percentage of patients who required intensive care units (ICU) care varied from 5% to 10% in Europe [29, 82]. Because COVID-19 treatment does not require ICU for a large number of patients, the former communist countries, which had better hospital facilities, were able to provide the necessary public health care for all patients. Even with regard to the needs of the ICU, Eastern European countries did not face the Spanish or Italian scenario, as they had sufficient primary and secondary care capacity to handle with the moderate and severe outcomes of COVID-19 [83].

There are some other factors which can be important for understanding the different initial outcomes between these countries. There is some speculation about protection from BCG vaccine, which is still mandatory in Eastern European countries [84]. Also, what requires caution in the analysis is the issue of testing, because fewer tests mean fewer identified cases [44].

However, for economic reasons, lockdown is not a long-term solution. Several studies evidenced that European countries will face socio-economic losses due to COVID-19 epidemics [85, 86] and that the consequences will be different in Eastern and Western European countries [87]. Hence, the Eastern European countries have yet to see the seriousness of COVID-19, as measures are relaxed and the capacity of the health system loses support in lockdowns and bans [88]. This is already obvious based on data from the end of July 2020, which show that COVID-19 is again very active in Serbia and some other neighbouring countries [89].

On the other hand, the Western European countries, although not rigid in the first response, will probably cope better with COVID-19 in the future. It is important to remember that this reading of the data is set around statistics on the initial outbreak and that a number of countries have yet to reach the peak of their outbreaks, as implied in reports from October 2020 [90].

Although our six-dimensional framework is not designed to be a forecasting tool in terms of the initial severity of the COVID-19 we can see that the chosen structural and process indicators gave a fair prediction of the actual outcome.

However, our study is not without limitations. The strongest one is about the data. This is especially true for outcome data (morbidity and mortality). As noted in several studies [91, 92], countries use incompatible methodologies for counting infected and deceased patients, which makes comparing data problematic. Moreover, newly established daily trackers of government measures and reduced mobility are still in the process of being redefined and established. Once the data limitation is overcome, we can focus our research on a more detailed analysis of the relationship between the expected and actual outcome. The choice of dimensions should also be revisited in the future. As we explained earlier, pandemics are complex situations and it takes time to collect all the pieces of the puzzle, especially if the analysis is to be expanded to other regions and a larger set of countries.

## Author Contributions

**Conceptualization:** Dalibor Petrović, Marijana Petrović, Nataša Bojković, Vladan P. Čokić.

**Data curation:** Dalibor Petrović, Marijana Petrović, Vladan P. Čokić.

**Formal analysis:** Dalibor Petrović, Marijana Petrović.

**Methodology:** Dalibor Petrović, Marijana Petrović, Nataša Bojković.

**Visualization:** Dalibor Petrović, Marijana Petrović.

**Writing – original draft:** Dalibor Petrović, Marijana Petrović, Nataša Bojković, Vladan P. Čokić.

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
