## [Decision Letter · Decision Letter 0]

8 Oct 2020

PONE-D-20-27597

An integrated view on society readiness and initial reaction to Covid–19: a study across European countries

PLOS ONE

Dear Dr. Petrovic,

Thank you for submitting your manuscript to PLOS ONE. After careful consideration, we feel that it has merit but does not fully meet PLOS ONE’s publication criteria as it currently stands. Therefore, we invite you to submit a revised version of the manuscript that addresses the points raised during the review process.

Please see comments by the reviewers.  In you revised manuscript, kindly provide point by point response to their queries.

We look forward to receiving your revised manuscript.

Kind regards,

Muhammad Adrish

Academic Editor

PLOS ONE

Journal Requirements:

2. Please upload a copy of Figure 4, to which you refer in your text on page 25. If the figure is no longer to be included as part of the submission please remove all reference to it within the text.

Reviewers' comments:

Reviewer's Responses to Questions

**Comments to the Author**

1. Is the manuscript technically sound, and do the data support the conclusions?

Reviewer #1: Yes

Reviewer #2: Yes

Reviewer #3: Yes

2. Has the statistical analysis been performed appropriately and rigorously? 

Reviewer #1: Yes

Reviewer #2: Yes

Reviewer #3: Yes

3. Have the authors made all data underlying the findings in their manuscript fully available?

Reviewer #1: No

Reviewer #2: Yes

Reviewer #3: Yes

4. Is the manuscript presented in an intelligible fashion and written in standard English?

Reviewer #1: Yes

Reviewer #2: Yes

Reviewer #3: Yes

5. Review Comments to the Author

Reviewer #1: I commend the efforts of the authors in contributing to literature on pandemics, especially COVID-19. However, I do have some concerns with some constructions of sentences. More editing and proofreading is needed, emphasisng on punctuations.

Introduction:

'This allowed us to more sophisticatedly analyse how European countries stand relative to each other in term of readiness and reaction to COVID-19'. Why is the word sophiscatedly' used in this sentence? I suggest 'to rigourously analyse'

'The second is whether the application of more rigorous measures can complement to the preparedness capacity of a country. Please delete 'to' in this sentence.

Materials and Methods:

'Namely this gave us the chance to bring the early outcome of pandemics (in terms of morbidity and mortality) together with readiness structure and reaction process'. What are you 'namely' here? I don't understand this sentence construction

'With the Covid-19 outbreak one of the key question was will well prepared health systems be resilient against the COVID-19 epidemic' Is this a statement or a question?

'Beside preparedness and trust one more important element of country reediness'. Readiness and not reediness. Please correct.

'Eventually, we encompass all dimensions and hypothesise that the obtained performance levels correspond the most to the actual outcome'. Plural od hypohtesis is hypotheses. Please correct.

Results:

'Low performance by the same criterion pushed Netherland and Germany down'. Netherlands

Discussion and conclusions:

'Our findings revile whole complexity of efforts to put Covid-19 outbreak under control'. Please check; revealed and not reviled, which means criticizing abusively or angrily.

'Unlike western European countries...' Should be Western.

'However, a better overall preparedness score for the Western health care systems does not necessarily mean that they are better equipped for fighting against viruses such a Covid-19.' Should read 'such as Covid-19.

'It is important to remind that this data reading is set around statistics on the initial outbreak and a number of countries are yet to reach the peak of their outbreaks'. 'to remember...'

Reviewer #2: Job well-done author but the methodology used was not clearly specific and outlined, pls outline the study methodology process. For example study designs, duration, population, eligibility, source, search strategy.

Pls mention the sampling strategy e.t.c and additional referencing to support some of your claim in the study, especially from line 530- 543. Thank you!

Reviewer #3: It is well written manuscript. A few grammatical errors were detected and a through review of sentences is suggested. A few relevant research articles are suggested which can further supplement the introduction and discussion section. They are as follows:

Kandel N, Chungong S, Omaar A, Xing J. Health security capacities in the context of COVID-19 outbreak: an analysis of International Health Regulations annual report data from 182 countries. The Lancet. 2020 Mar 18.

Tangcharoensathien V, Calleja N, Nguyen T, Purnat T, D’Agostino M, Garcia-Saiso S, Landry M, Rashidian A, Hamilton C, AbdAllah A, Ghiga I. Framework for managing the COVID-19 infodemic: methods and results of an online, crowdsourced WHO technical consultation. Journal of medical Internet research. 2020;22(6):e19659.

Kraus S, Clauss T, Breier M, Gast J, Zardini A, Tiberius V. The economics of COVID-19: initial empirical evidence on how family firms in five European countries cope with the corona crisis. International Journal of Entrepreneurial Behavior & Research. 2020 May 26.

6. PLOS authors have the option to publish the peer review history of their article (what does this mean?). If published, this will include your full peer review and any attached files.

Reviewer #1: No

Reviewer #2: No

Reviewer #3: No

---

## [Author Response · Author response to Decision Letter 0]

19 Oct 2020

Please see attached Response to Reviews document.

---

## [Decision Letter · Decision Letter 1]

11 Nov 2020

An integrated view on society readiness and initial reaction to COVID–19: a study across European countries

PONE-D-20-27597R1

Dear Dr. Petrovic,

We’re pleased to inform you that your manuscript has been judged scientifically suitable for publication and will be formally accepted for publication once it meets all outstanding technical requirements.

Kind regards,

Muhammad Adrish

Academic Editor

PLOS ONE

Additional Editor Comments (optional):

Reviewers' comments:

Reviewer's Responses to Questions

**Comments to the Author**

1. If the authors have adequately addressed your comments raised in a previous round of review and you feel that this manuscript is now acceptable for publication, you may indicate that here to bypass the “Comments to the Author” section, enter your conflict of interest statement in the “Confidential to Editor” section, and submit your "Accept" recommendation.

Reviewer #1: All comments have been addressed

Reviewer #3: All comments have been addressed

2. Is the manuscript technically sound, and do the data support the conclusions?

Reviewer #1: Yes

Reviewer #3: Yes

3. Has the statistical analysis been performed appropriately and rigorously? 

Reviewer #1: Yes

Reviewer #3: Yes

4. Have the authors made all data underlying the findings in their manuscript fully available?

Reviewer #1: Yes

Reviewer #3: Yes

5. Is the manuscript presented in an intelligible fashion and written in standard English?

Reviewer #1: Yes

Reviewer #3: Yes

6. Review Comments to the Author

Reviewer #1: (No Response)

Reviewer #3: Thank you for addressing all the points highlighted in the previous review. The manuscript is now suitable for publication.

7. PLOS authors have the option to publish the peer review history of their article (what does this mean?). If published, this will include your full peer review and any attached files.

Reviewer #1: No

Reviewer #3: No

---

## [Editor Report · Acceptance letter]

13 Nov 2020

PONE-D-20-27597R1 

An integrated view on society readiness and initial reaction to COVID–19: A study across European countries 

Dear Dr. Petrovic:

I'm pleased to inform you that your manuscript has been deemed suitable for publication in PLOS ONE. Congratulations! Your manuscript is now with our production department. 

Kind regards, 

on behalf of

Dr. Muhammad Adrish 

Academic Editor

PLOS ONE